# ZeroSumEval: Scaling LLM Evaluation with Inter-Model Competition

## Abstract

Evaluating the capabilities of Foundation Models has traditionally relied on static benchmark datasets, human assessments, or model-based evaluations — methods that often suffer from overfitting, high costs, and biases. We introduce ZeroSumEval, a novel competition-based evaluation protocol that leverages zero-sum games to assess LLMs with dynamic benchmarks that resist saturation. ZeroSumEval encompasses a diverse suite of games, including security challenges (Capture the Flag), classic board games (chess), and knowledge tests (MathQuiz). These games are designed to evaluate a range of AI capabilities such as strategic reasoning, planning, knowledge application, safety, and adaptability. A key novelty is integrating automatic prompt optimization to ensure fair comparisons by eliminating biases from human prompt engineering and support arbitrary prompting strategies. Furthermore, ZeroSumEval measures AI models' abilities to self-improve from limited observations and assesses their robustness against adversarial or misleading examples during prompt optimization. Building upon recent studies that highlight the effectiveness of game-based evaluations for LLMs, ZeroSumEval enhances these approaches by providing a standardized and extensible framework for rigorous assessment. We find ZeroSumEval correlates strongly with expensive human evaluations (Chatbot Arena) and disagrees with benchmarks with known overfitting and saturation issues. Inspecting match traces reveals models that allocate more tokens to thought processes perform strongly in games involving planning capabilities.

## 1 Introduction

Large Language Models (LLMs) are being developed at an unprecedented pace (Zhao et al., 2024), requiring significant investment for their training and refinement (Kevin Lee, 2024; Miller, 2022; Kimball, 2024). As the performance and complexity of these models continue to grow (Chen et al., 2024b), selecting the most appropriate model for a specific application has become an increasingly challenging and costly decision(Kaplan et al., 2020; Hoffmann et al., 2022). Benchmarking emerges as a critical tool in this context (Laskar et al., 2023; Qin et al., 2023), providing standardized metrics and evaluations to guide these choices.

With the rapid growth of generative technologies built on top of Large Language Models (OpenAI, 2022; Google, 2024; Anthropic, 2024b; Ormazabal et al., 2024; Mistral, 2024; Dubey et al., 2024a; Yang et al., 2024), it has been increasingly difficult to evaluate these models comprehensively (Guo et al., 2023). Current benchmarking practices face several significant issues. Many benchmarks suffer from data contamination (Yang et al., 2023), where models inadvertently train on portions of the test data (Dubey et al., 2024a; Groeneveld et al., 2024), leading to inflated performance metrics. Sensitivity to prompt variations (Alzahrani et al., 2024b) and a lack of diversity in evaluation tasks (Laskar et al., 2024) further undermine the reliability and robustness of these benchmarks. Additionally, the high cost and effort required to develop new benchmarks often result in outdated evaluation methods that do not keep pace with the rapid development of LLMs (Kiela et al., 2021; Vu et al., 2023).

An observed disparity exists between the computational resources measured in floating-point operations per second, or FLOPs used to train LLMs and those allocated for their evaluation. Training these models involves massive computational efforts (Hoffmann et al., 2022), yet the evaluation

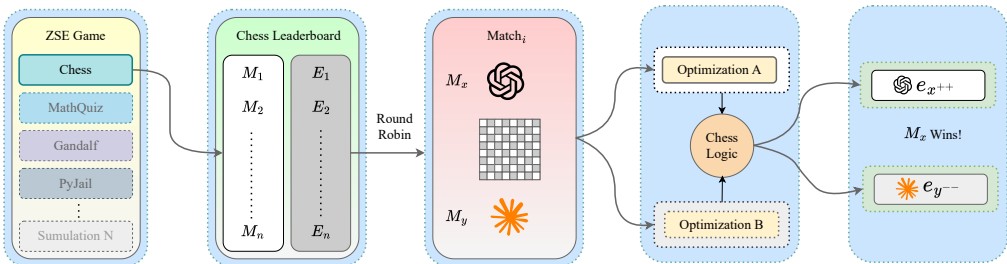

Figure 1: The ZeroSumEval suite of benchmarks provides dynamic simulations with head to head model competition to create robust and scalable model evaluations and leaderboards. Integrated automatic prompt optimization minimizes biases introduced by prompting and hand-engineering.

phase typically utilizes a negligible fraction of this capacity (Laskar et al., 2024). Scaling up evaluation by increasing the number of evaluation tokens is essential for a more thorough understanding of model capabilities. Traditionally, this scaling involves incorporating human-crafted independent and identically distributed (i.i.d.) data (Holland et al., 2018), which is resource-intensive (Hutchinson et al., 2021) and may not adequately capture the complexities of language (Mehrabi et al., 2021) and reasoning required to challenge advanced LLMs (Gudibande et al., 2023) or even LLM generated (Karpinska et al., 2021).

Previous work has proposed the use of games as benchmarks (Topsakal et al., 2024), offering a promising avenue for evaluating complex reasoning (Wong et al., 2023) and decision-making abilities of LLMs (Warstadt et al., 2023; Park et al., 2023; Wang et al., 2023). Games provide interactive and dynamic environments that can test models beyond static datasets. However, existing game-based benchmarks are often *(i)* inflexible and limited in scope, *(ii)* not easily extendable, *(iii)* restricted in their effectiveness for comprehensive model evaluation, and *(iv)* depend on predefined prompts.

Scaling evaluation is fundamental not only for assessing performance but also for uncovering hidden dynamics within LLMs, such as potential backdoors or biases (Schuster et al., 2020), and for evaluating their emerging reasoning capabilities (Brown et al., 2020; Sanh et al., 2022; Wei et al., 2023b;a). Implementing environments for simulations or games offers a scalable solution to these challenges (OpenAI et al., 2019; OpenAI, 2019; Silver et al., 2016; 2017; Zheng et al., 2021).

Existing evaluation protocols possess several key issues:

*(i)* **Prompt Sensitivity**: Previous work (Zheng et al., 2024; Pezeshkpour & Hruschka, 2023; Lu et al., 2022; Alzahrani et al., 2024a; Wang et al., 2024a) has shown that models are sensitive to benchmark formats. By sheer chance, a model could be presented with a prompting method that's either favorable or detrimental. These prompt modifications are shown to result in substantially different relative performance between models (Alzahrani et al., 2024a). By testing models in varied scenarios within a controlled environment, we can assess and improve their robustness to different prompts. Crucially, different models are not optimized for the same prompts due to variations in data mixtures and algorithmic implementations. Using identical prompts across all models may therefore lead to unfair comparisons.

*(ii)* **Limited Diversity**: Traditional evaluation methods often rely on static datasets, which are inherently limited by their dependency on human curation and annotation. This makes it challenging to continuously introduce new, diverse test data. An extensible simulated environment, however, allows for a wide array of dynamically generated games and scenarios, enhancing the diversity and scalability of evaluation tasks.

*(iii)* **Extensibility**: Once established, the environment can be easily expanded to include new games, rules, and scenarios, facilitating continuous evaluation improvements.

*(iv)* **Crowd and Annotator Bias**: LLM evaluations conducted by large crowds often tend to be susceptible to social hacking, and it can depend on geographic, temporal, and narrative factors Gururan-

gan et al. (2018). Controlled and interpretable environments can mitigate these biases by providing consistent, objective evaluation criteria.

*(v)* **Saturation**: With the rapid improvement of LLMs, evaluation benchmarks quickly become obsolete and saturated, with frontier models achieving almost perfect scores, which necessitates the development of new benchmarks. On the opposite extreme, benchmarks that are too difficult would result in almost random scores. Both extremes result in a lack of granularity to distinguish models. Therefore, benchmarks posing moderate difficulty to frontier models will need to be continuously developed as models improve. For instance, GSM8K (Cobbe et al., 2021) tests models on grade school-level math, and most state-of-the-art models achieve scores above 90% (Dubey et al., 2024b; Anthropic, 2024a). Thus, the more difficult MATH (Hendrycks et al., 2021b) dataset, which consists of math competition questions, was developed and is now commonly used[1]. A similar trend is observed in academic examination benchmarks with the migration from MMLU (Hendrycks et al., 2020) to MMLU-Pro (Wang et al., 2024b) and GPQA. (Rein et al., 2023)[2].

To address these challenges, we introduce ZEROSUMEVAL, a flexible and extensible open-source framework designed to scale LLM evaluation through the simulation of two-player zero-sum games. Our framework allows for comprehensive and robust assessment by providing models with multiple opportunities to make legal moves, thereby accommodating occasional errors and offering a more nuanced understanding of their capabilities.

1. **Scaling Evaluation by Simulation**: We demonstrate how simulation environments can effectively scale the evaluation process.

2. **Flexible and Extensible Framework**: ZEROSUMEVAL is designed to be adaptable, allowing researchers and practitioners to customize and extend the evaluation environment to suit diverse needs.

3. **Robustness to Prompt Sensitivity**: By incorporating automatic prompt optimization, our framework mitigates issues related to prompt sensitivity, leading to more reliable evaluation outcomes.

4. **Enhanced Interpretability**: The structured environment facilitates easier interpretation of model behaviors, aiding in the identification of strengths and weaknesses.

5. **Error Accommodation**: Models are given multiple chances to make legal moves, ensuring that occasional missteps due to inherent stochasticity do not disproportionately affect the overall evaluation.

## 2 RELATED WORK

### 2.1 STATIC LLM BENCHMARKS

Until recently, LLMs were evaluated on Natural Language Understanding (NLU) tasks from benchmark collections like GLUE (Wang, 2018) and SuperGLUE (Wang et al., 2019), which included tasks like paraphrase classification and sentiment analysis. As LLMs developed, they acquired emergent capabilities beyond generating plausible text, such as reasoning, generating code, and instruction following (Brown et al., 2020; Wei et al., 2022). With these newly found capabilities, new benchmarks were developed to quantify these abilities. As models improve, more difficult benchmarks are created. For example:

- **Reasoning:** undergraduate level academic questions are tested via MMLU (Hendrycks et al., 2020), while GPQA (Rein et al., 2023) tests models with graduate level questions. All aforementioned benchmarks score models based on the likelihood of specific tokens for the answer keys in a multiple-choice setting.

- **Mathematics:** GSM8K (Cobbe et al., 2021) evaluates models on elementary level arithmetic, while MATH (Hendrycks et al., 2021b) tests on competition level mathematics. Both benchmarks evaluate the model in a few-shot setting by encouraging models to output chains of thought followed by the numeric answer in a specific format.

---

[1] HuggingFace's Open LLM Leaderboard (Beeching et al., 2023; Fourrier et al., 2024) migrated from GSM8K in v1 to MATH in v2.

[2] Similar to 1, the leaderboard transitioned from MMLU in v1 to MMLU-Pro and GPQA in v2.

- **Coding:** HumanEval (Chen et al., 2021) test models on basic coding, while APPS (Hendrycks et al., 2021a) uses coding competition questions. These benchmarks generate Python code by prompting LLMs with function docstrings or written specifications, and run input/output test-cases on the generated code.

Critisism of these types of static benchmarks are outlined in Section 1.

## 2.2 Comparative LLM Benchmarks

**LLM Game Evaluations** To address the static benchmark issues highlighted in Section 1, the paradigm of evaluating agentic capabilities through simulations has been applied successfully in multiple prior works. Evaluation frameworks comprising multiple games include: *(i) ChatArena* (Wu et al., 2023), which includes Chess, Tic-Tac-Toe, Rock-Paper-Scissors, and others, *(ii) GridGames* (Topsakal et al., 2024), implementing Tic-Tac-Toe, Connect Four, and Gomoku, and *(iii) GameBench* (Costarelli et al., 2024), which is the most diverse, as they developed 9 games, include non-deterministic and imperfect information games.

**Limitations of LLM Game Evaluations** All the aforementioned benchmark frameworks are implemented with manually written prompts for all models, and sometime suggest a strategy within the prompt, such as ChatArena prompting models to output a random move in Rock-Paper-Scissors. GameBench tries to optimize model results by utilizing two prompting strategies: *(i)* Chain of Though (CoT), and *(ii)* Reasoning via Planning (RAP), but the issue of static prompt still persists. This could explain the poor performances they observed, such as GPT-4 achieving almost random results on some tasks.

**Comparative Human Evaluations** A popular head-to-head LLM evaluation framework is Chatbot Arena[3] (Chiang et al., 2024), which allows users to prompt two anonymous LLMs with arbitrary prompts and to vote for the better response. This creates a diverse evaluation that effectively ranks all models in a leaderboard. However, it suffers from two issues: *(i)* human evaluations are slow and laborious, and adding new models requires prolonged evaluation periods until sufficient votes are acquired for a confident placement, and *(ii)* human evaluations contain human biases, such as prompt over-representation (Dunlap et al., 2024) and bias to verbose and "pretty" responses (Chen et al., 2024a; Park et al., 2024; Li et al., 2024).

## 3 Methodology

In this section, we describe the technical details of ZeroSumEval including design choices, the importance of automatic prompt optimization, and game selection/categorization. At its core, ZeroSumEval provides controlled environments to observe models competing against each other to win competitive games. In particular, ZSE controls (i) the role and information each model has access to at any point in the simulation and (ii) the data models can use to optimize/modify their own prompts.

### 3.1 Capabilities

The games within ZSE are designed to evaluate specific capabilities in a controlled environment:

**Reasoning** Board games and cybersecurity scenarios require models to perform complex, multi-step reasoning. They test the models' ability to process information, predict outcomes, and formulate strategies in dynamically changing environments.

**Planning** Board games also involve long-term strategy, requiring models to anticipate the consequences of their actions several moves ahead. This assesses the model's foresight, adaptability, and capacity for nuanced decision-making.

---

[3]formerly LMSYS, not to be confused with ChatArena.

**Knowledge Application**   Models must recall and apply mathematical knowledge to solve problems in question answering type games. This setup provides a direct assessment of the models' ability to retrieve, interpret, and implement factual information in structured problem-solving.

**Creativity**   Models successful at cybersecurity type games must exhibit creativity to successfully create secure environments and break them.

## 3.2   GAME DESIGN

ZEROSUMEVAL supports an expanding suite of game types designed to test the aspects of LLM performance described above. The mix we showcase includes both well-known and established games, such as chess, as well as more special-purpose games (e.g. MathQuiz). For completeness and reproducibility, we describe the implementations of MathQuiz and PyJail. The following set of games are selected to encompass a range of cognitive capabilities, including strategic reasoning, planning, knowledge application, and creativity:

**Board Games (Chess)**   Classic board games like chess serve as a benchmark for strategic reasoning and long-term planning. They require models to engage in multi-step thinking, manage trade-offs, and foresee opponent moves. This category is instrumental in evaluating a model's ability to plan several moves ahead, adapt its strategies, and make complex decisions under uncertainty[4].

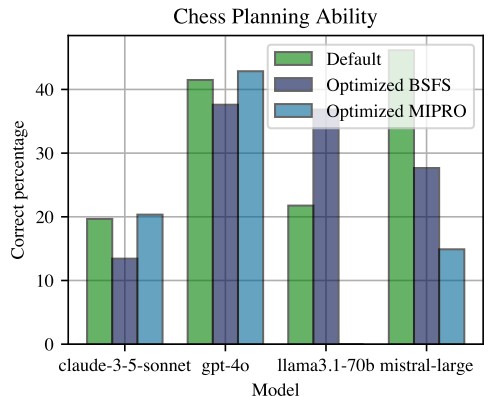

Figure 2: The effect of prompt optimization on the proportion of correct moves. Moves are classified as correct if the evaluation, as determined by Stockfish 17 (The Stockfish Developers, 2024) with depth 15, does not decrease by more than 0.3 points (pawn equivalent). Models react differently to prompts and have varying prompt optimization abilities.

**Question-Answer   Games   (MathQuiz)**   These games are constructed to measure models' knowledge recall and logical reasoning abilities. MathQuiz, for instance, challenges models to both create and answer arithmetic and mathematical questions, assessing their understanding of mathematical concepts, computational accuracy, and step-by-step problem-solving skills. Our implementation of MathQuiz tasks a teacher player to create a challenging math problem and prove that the problem is valid and solvable. A student player then attempts to answer the generated math problem. The student wins the game by answering the question correctly or if the teacher fails to create a valid question.

**Cybersecurity Games (PyJail)**   PyJail involves python "capture the flag" cybersecurity challenges, targeting the model's ability to create puzzles and interact with a restricted python environment to strategize solutions. The PyJail game is structured into three stages. The first statically parses a player generated PyJail program to provide feedback on the syntax and semantic structure. Given validity, the challenge code is inserted into the environment, and the same player model must commit a solution that is tested dynamically to prove the challenge's feasibility. A unique flag is stored in the target variable at runtime, which prevents any trivial method to cheat the challenge. The second player will complete the same step, provided a restricted view of the environment and limited context. The game ends if first player is unable to create a valid challenge or the flag is retrieved by the attacker.

---

[4]Chess has a rich history as a testbed for strategy and planning. See https://github.com/carlini/chess-llm and https://huggingface.co/spaces/mlabonne/chessllm for examples of LLMs playing chess.

## 3.3 SCALABLE VERIFICATION

The MathQuiz and PyJail games require competing models to generate complex challenge environments and solutions. Since verification of the knowledge-based challenges by a human in the loop is not scalable, we design a method to verify model output using an automated manager in a two-fold generation and verification process. This is accomplished by defining a target outcome (e.g., the answer to a math question or a CTF flag) as the basis for verifying generated input, and regulating the model context at each stage.

The exact process (illustrated in Figure 3) is outlined as follows:

(i) The generator model receives a target and attempts to output a valid challenge that resolves to the specific target.

(ii) In the verification step, the manager restricts the model's context to ensure no direct access to the target, and asks the generator model to solve the previously generated challenge.

(iii) If the manager determines the verification is successful (by matching the target with the generator's solution), the game proceeds. Otherwise, the generator model is deemed to have failed to generate a valid challenge.

Figure 3: State diagram of the verification process involving the Game Manager and the Generator. Blue boxes indicate deterministic steps and green boxes indicate steps involving the model.

This method ensures the generated challenge environment is valid and a solution is proven possible by the generator. The design also correctly penalizes models that directly generate memorized questions as it is likely to have been memorized by other models, thereby encouraging models to create challenging and novel questions. Finally, the scalability of the evaluation is preserved as the capabilities of models scale.

## 3.4 AUTOMATIC PROMPTING

Automatic prompting is an essential component of the ZEROSUMEVAL framework for several reasons. First, it allows models to learn to play new games through self-optimization, demonstrating their ability to adapt to different scenarios without human intervention. Second, it removes the human element in prompt engineering, thereby reducing biases and variations introduced by manual prompt construction (Zheng et al., 2024; Pezeshkpour & Hruschka, 2023; Alzahrani et al., 2024a). Third, automatic prompting serves as a measure of a model's ability to self-improve at inference time, providing insight into its adaptability and strategic reasoning skills.

We leverage the DSPy (Khattab et al., 2023) approach to implement automatic prompt optimization in our framework. DSPy allows models to autonomously explore and select optimal prompts based on the current game context, dynamically adjusting strategies to maximize performance. We also make use of DSPy Assertions (Singhvi et al., 2024) to simulate interactivity between the models and the game environment by allowing a number of retries (with feedback from the game) when the model makes an invalid move. Although we find DSPy has the flexibility and generalizability to support various models and games, ZEROSUMEVAL supports alternative automatic prompt optimization techniques if required.

Through prompt optimization, models can develop improved strategies as they encounter diverse game scenarios. For example, in a chess game, models equipped with optimized prompting demonstrated a higher proportion of correct moves compared to their counterparts using default prompts Figure 2. This not only reveals the models' enhanced strategic reasoning but also emphasizes the significance of prompt optimization in robust performance evaluation.

By incorporating automatic prompting, ZEROSUMEVAL addresses benchmark sensitivity. The prompt optimization integrates game validation mechanisms into the optimization process, allowing models to observe tangible outcomes and refine their prompt strategies. Consequently, this mitigates the variations in performance due to prompt sensitivity, leading to a more consistent and reliable evaluation of model capabilities.

**Datasets and Optimizers**   To perform the automatic prompt optimization process, models require examples of gameplay (inputs and outputs) and prompt optimizers. We create standard datasets manually for each game available to all models for the optimization. The available datasets are described in Table 1. Through DSPy, ZEROSUMEVAL supports multiple types of optimizers. In this work, we focus on (i) BootstrapFewShot (ii) BootstrapFewShotRandomSearch (Khattab et al., 2023) and (iii) MIPROv2 (Opsahl-Ong et al., 2024).

| Dataset | Source | Description |
| --- | --- | --- |
| chess_stockfish | conacts/stockfish_dataset[5] | stockfish vs stockfish games |
| chess_puzzles | (Schwarzschild et al., 2021) | chess puzzles. |
| mathquiz_gsm8k | (Cobbe et al., 2021) | grade school level math QA |
| mathquiz_hendrycks_math | (Hendrycks et al., 2021b) | advanced math QA |
| pyjail_ctf_llm | (Shao et al., 2024) | Pyjail style Capture The Flags (CTFs). |

Table 1: Overview of datasets used in the evaluation framework.

An interesting direction out of the scope of this work is enabling models to learn games via self-play. This would reduce manual effort needed to create new games for ZEROSUMEVAL and measure a model's ability to effectively explore a space without supervision.

## 3.5   RATINGS

ZEROSUMEVAL utilizes an easily computable rating system derived from the outcomes of competitive games between models. Each model receives a rating based on its win-loss record over multiple games, allowing for a rapid and scalable oversight of model capabilities. This framework seamlessly incorporates new games, providing continuous and dynamic evaluation as models improve.

Following recent suggestions for LLM rating systems by Boubdir et al. (2023); Chiang et al. (2023), we employ the Bradley-Terry (BT) rating system, an alternative to the Elo system, to rate models. The BT model is permutation-invariant and assumes a fixed win rate for each model pair, maximizing the likelihood of observed outcomes (Bradley & Terry, 1952). This choice is more suitable than the traditional Elo system, which was designed for human chess players with varying skill levels, whereas LLMs have fixed skill levels defined by their weights (Elo, 1967).

ZEROSUMEVAL's rating system facilitates analysis of model behaviors. It allows us to observe not only the relative strategic planning capabilities of models but also their capacity for self-improvement through prompt optimization. For instance, analysis of models' gameplay strategies in chess revealed that prompt-optimized models allocate more reasoning words in their decision-making process, suggesting a deeper level of planning (Figure 4).

## 4   EXPERIMENTS

In this section, we describe the experiments to demonstrate the effectiveness of the ZEROSUMEVAL as a dynamic leaderboard. We also design experiments to evaluate the effect of prompt optimization on the performance of various large language models (LLMs) under various simulations.

### 4.1   MODEL SELECTION AND EXPERIMENTAL SETUP

We select four models of varying sizes and capabilities for this study: GPT-4o, Claude 3.5 Sonnet, LLaMA 3.1-70B-Instruct, and Mistral-Large. These models represent a range of architectures and training scales, providing a diverse set for evaluating the generalizability of the ZEROSUMEVAL framework.

The experiments involve running a multiple round-robin tournaments to simulate competitive game-play among the model (50-100 games per experiment). In addition to measuring model performance on the games in the ZEROSUMEVAL suite, we also examine how the models' performance changes with different prompt optimization techniques. Each tournament round involves all possible match permutations between model variants, after which the models' ratings are calculated using the Bradley-Terry model (Bradley & Terry, 1952). The primary goal of this ablation study is to assess each model's responsiveness to the optimization process and to identify resulting behavioral changes.

For the automatic prompt optimization, we utilize three optimizers commonly used in DSPy: BootstrapFewshot (BSFS), BootstrapFewshotWithRandomSearch (BSFSRS), and MIPROv2, targeting the ChainOfThought module in DSPy.

## 4.2 GAMES FOR ANALYSIS

Although ZEROSUMEVAL supports a range of games for assessing different capabilities, our detailed set of experiments focus primarily on Chess to analyze the models' planning abilities. This decision is motivated by the interpretability of Chess gameplay and its complexity, which provides an ideal testbed for assessing strategic reasoning and decision-making.

## 5 RESULTS

### 5.1 RATINGS AND PERFORMANCE TRENDS

Table 2 provides the ratings for each model variant across the games. The results indicate GPT-4o and Claude 3.5 Sonnet typically perform best with GPT-4o slightly ahead. This agrees with leaderboards based on human ratings, such as ChatbotArena (Chiang et al., 2024).

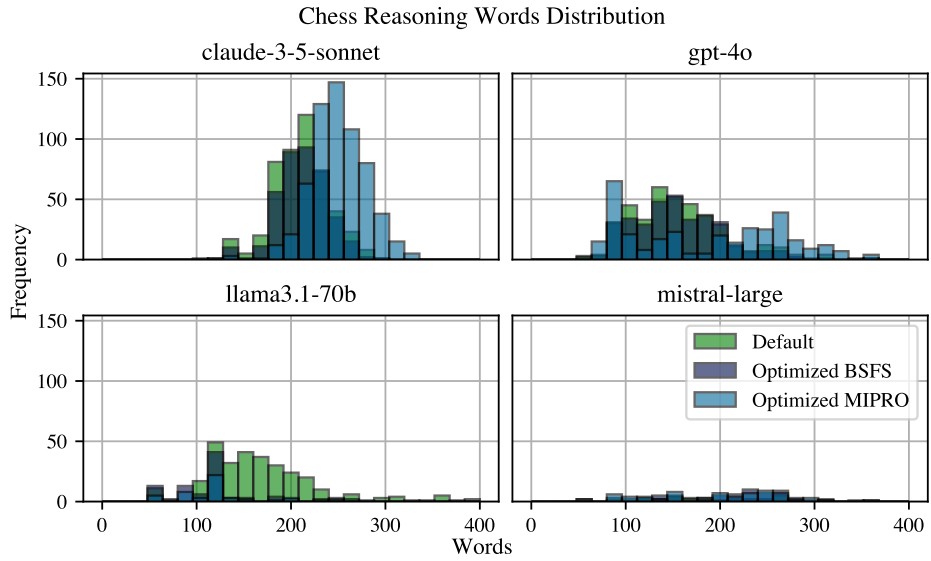

Figure 4: The distribution of CoT words used for each model and prompt optimization technique. In general, prompt optimized models spend more words reasoning than their non-optimized counterparts, especially with MIPROv2 optimization.

### 5.2 IMPACT OF PROMPT OPTIMIZATION ON PERFORMANCE

The experimental results (Table 3) reveal significant variations in model performance as a result of prompt optimization. Prompt optimization can even flip ranking as is the case with MIPROv2 -

highlighting the significant effect of prompt sensitivity. Prompt-optimized models typically exhibit improved strategic reasoning, as evidenced by an increased number of correct moves and a more favorable distribution of move evaluations (Figure 2). ZEROSUMEVAL provides the capability to compare models across prompt optimization strategies, leading to fairer evaluations and more robust leaderboards.

Figure 4 illustrates the distribution of CoT words for each model with different prompt optimization techniques. Notably, models optimized using MIPROv2 demonstrate a tendency to allocate more words to their reasoning process compared to their default counterparts, suggesting deeper planning and strategic consideration.

| Model | Chess (MIPRO) | MathQuiz (Default) | PyJail (Default) |
|---|---|---|---|
| GPT-4o | 1202.97 | 1048.12 | 1025.58 |
| Claude 3.5 Sonnet | 1000.00 | 962.51 | 1017.17 |
| Mistral-Large | 940.88 | 982.85 | 1000.00 |
| LLaMA 3.1 70B | 856.15 | 1006.52 | 953.15 |

Table 2: Performance ratings of various models across different tasks. The ratings are computed using the MIPRO-optimized approach for the Chess task and default settings for MathQuiz and PyJail tasks.

| Model | Default | BSFS | BSFSRS | MIPRO |
|---|---|---|---|---|
| | Rating (CI) | Rating (CI) | Rating (CI) | Rating (CI) |
| Claude 3.5 Sonnet | **1028** (890-1153) | 1000 (871-1126) | 1000 (862-1147) | 984 (837-1060) |
| Mistral-Large | 942 (889-1005) | 1016 (963-1073) | 1014 (952-1069) | **1023** (965-1090) |
| LLaMA 3.1 70B | 978 (918-1054) | 951 (888-1039) | 1030 (962-1089) | **1035** (967-1107) |
| GPT-4o | 962 (880-1034) | 1016 (909-1101) | 966 (874-1044) | **1055** (987-1133) |

Table 3: Results of engaging each model in competition against itself optimized by our choices of optimizers. We can see that for Mistral, LLaMA, and GPT-4o, MIPRO outperforms all other optimizers. It is interestingly not the case with Claude. Ratings are shown with their 95% confidence intervals (CI). The highest rating for each model is in bold.

## 6 CONCLUSION

The dynamic, competitive nature of ZEROSUMEVAL's evaluation provides a more robust and trustworthy measurement of AI model capabilities, advancing the state of benchmarking in large language models. By leveraging zero-sum games, we ensure that models are consistently challenged with diverse, evolving tasks, minimizing the risk of overfitting and saturation commonly observed in static benchmarks. Additionally, the integration of automatic prompt optimization offers a more holistic evaluation framework that captures not only a model's static performance but also its dynamic capacity for self-improvement.

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

## A  APPENDIX

Table 4: Exact model versions used in our evaluations.

| Model | Version |
|-------|---------|
| GPT-4o | `gpt-4o-2024-08-06` |
| Claude 3.5 Sonnet | `claude-3-5-sonnet-20240620` |
| Mistral-Large | `mistralai/Mistral-Large-Instruct-2407` |
| Llama 3.1 70B | `meta-llama/Meta-Llama-3.1-70B-Instruct` |

