# OpenReview forum: "ZeroSumEval: Scaling LLM Evaluation with Inter-Model Competition"
_ICLR.cc/2025/Conference — ICLR 2025 Conference Withdrawn Submission_

### Official Review · Reviewer_PUfS · 2024-10-28

**Soundness:** 1
**Presentation:** 2
**Contribution:** 2
**Rating:** 3
**Confidence:** 3

**Summary:**

The paper proposes ZeroSumEval: a framework / suite of zero-sum games / tasks that may be used for evaluating planning/reasoning capabilities of LLMs. The authors argue that it is key points of this framework are:

1. Inclusion of automatic prompt optimization, to reduce sensitivity of models to prompts.
2. Designing the framework to be flexible and extensible.
3. Facilitating interpretation of what models are doing.
4. Allowing the evaluation process to scale (e.g., with increasingly difficult tasks) as the capabilities of models scale up

**Strengths:**

Important, interesting and topical area of research.

**Weaknesses:**

For some explicitly claimed contributions at the end of the Introduction, I don't really see how they are supported by any of the work described in the paper:

1. "We demonstrate how simulation environments can effectively scale the evaluation process" --> where / how do you show this in the paper?
2. "ZeroSumEval is designed to be adaptable" --> where in the paper do you explain how its design ensures adaptability/extensibility? I don't see anything about even a common/unified, well-defined and thought-out API (e.g., Gym-style API) that all tasks should implement. This is the minimum I would expect for a framework designed to make it easy to extend with new tasks.
3. "The structured environment facilitates easier interpretation of model behaviors" --> I don't fully get what "structured" means here, but regardless: I don't see how any of the design of the framework specifically facilitates interpretation of models or their behaviors. There is one small experiment in the paper trying to interpret something about the models, but this could've been done in exactly the same way outside the framework too.

---

I am also missing a discussion of an important area of related work. Prior to LLMs becoming a topic of research in the last few years, there have been multiple decades of research on other forms of AI for planning/reasoning tasks, including games. Many frameworks and evaluation protocols and competitions and benchmarks and everything have been created in these communities. Think of the General Game Playing community, for instance, and the frameworks and benchmarks they have developed. Many of these have explicitly tried to tackle some of the challenges the authors also point out in this paper, such as the question of how to develop a framework that can be easily extended with new tasks (new games). Various systems with game description languages come to mind. Why is none of this work discussed, and no inspiration drawn from it?

---

Other comments:
- Line 361 says that the *rating system* "facilitates analysis of model behaviors". I fail to see how the particular rating system used plays any role in this. Couldn't exactly the same kind of interpretation/analysis be done with any other rating system?
- Table 2: it would be nice to also include CIs here, like you do for Table 3.
- The (Dubey et al., 2024) reference is included twice in the list of references.

**Questions:**

1. "We demonstrate how simulation environments can effectively scale the evaluation process" --> where / how do you show this in the paper?
2. "ZeroSumEval is designed to be adaptable" --> where in the paper do you explain how its design ensures adaptability/extensibility?
3. "The structured environment facilitates easier interpretation of model behaviors" --> I don't fully get what "structured" means here, but regardless: I don't see how any of the design of the framework specifically facilitates interpretation of models or their behaviors?

---

### Official Review · Reviewer_vpxP · 2024-11-03

**Soundness:** 1
**Presentation:** 1
**Contribution:** 1
**Rating:** 3
**Confidence:** 3

**Summary:**

The authors propose ZeroSumEval, a benchmarking framework that evaluates LLMs by having them play against each other in 2 player zero sum games.

ZeroSumEval includes three games: Chess, MathQuiz and PyJail, with the possibility to add new games in the future. Each game is accompanied by a dataset (e.g. Stockfish games for chess or GSM8K for MathQuiz) on which it is possible to optimize LLM prompts before using those prompts when playing the game. Adding new games would therefore require also adding a new corresponding dataset.

By optimizing prompts on a fixed dataset before playing the game, the authors claim to mitigate the prompt sensitivity prevalent in benchmarks with fixed prompts.

**Strengths:**

The idea of using zero-sum games to create AI benchmarks that are robust to reward-hacking and difficult to saturate is a promising direction. MathQuiz and PyJail, two of the three games proposed in ZeroSumEval, have the potential to be good benchmarks if the authors can demonstrate that they lead to creating grounded problems that can be formally verified to be solvable.

The choice of LLMs for evaluation is also more than adequate.

**Weaknesses:**

Unfortunately, the paper suffers from major weaknesses which prevent me from recommending it for acceptance.

**1. Prompt optimization is biased**

A lot of the narrative of the paper revolves around "_integrating automatic prompt optimization to ensure fair comparisons by eliminating biases from human prompt engineering_". However, the experiments presented support the conclusion that the choice of optimizer greatly influences the relative performance of different LLMs. Notably, In Figure 2, optimization with BSFS hurts all models except LLama. MIPRO moderately helps Claude and GPT-4o, but greatly hurts Mistral (no result provided for Llama, or Llama gets 0% accuracy). The same holds for Table 3, where the authors comment that "_prompt optimization can even flip ranking as is the case with MIPROv2_".

As a result, it is unclear to me how the authors can claim that prompt optimization can reduce bias in evaluations. Instead, a more sound approach would be to consider each model-prompt pair as its own AI agent, and evaluate their performance accordingly.

**2. Paper lacks technical details**

The paper is generally vague, lacking technical details throughout. Phrasing is also often convoluted, which hurts comprehension. Among other things:

- Few details on MathQuiz and PyJail. Given those are games created specifically for this benchmark, as far as I can tell, I expect more details on the data distribution, the number of trials, the game dynamics, how LLMs are prompted (e.g. single-turn vs multi-turn), etc.
- Scalable verification: few details on how verification works in practice, e.g. on MathQuiz. Is the Manager operated by an LLM or simply a rule-based system? Is it general, or does any new game of this type require the implementation of a new verification mechanism?
- "_The prompt optimization integrates game validation mechanisms into the optimization process, allowing models to observe tangible outcomes and refine their prompt strategies._" What does this mean?
- The authors choose to report Bradley-Terry ratings. However, since it is less known, it would be useful to provide some details for interpretability. For instance, is a rating of 1200 much higher than 1000?

**3. No measure of statistical significance**

No standard error or confidence intervals are provided in the paper, except for Table 3. In Table 3, none of the the results are significant.

**4. Requirement for standard datasets**

Providing standard optimization datasets for each game and encouraging prompt optimization on those datasets strikes me as an odd choice, that is poorly motivated. This makes it hard to expand the benchmark, since adding new games also requires adding new datasets. It is also in a way inducing data contamination, since each LLM is provided with strong examples which it could incorporate into its prompt at the prompt optimization phase.

Even in the context of prompt optimization, why not perform prompt optimization directly on the game (in self-play) rather than on a fixed dataset?

I would like if the authors could better explain this design choice, since for me it seems to hurt the quality of the benchmark.

**Questions:**

See questions in the "weaknesses" section

---

### Official Review · Reviewer_2i9p · 2024-11-04

**Soundness:** 1
**Presentation:** 3
**Contribution:** 1
**Rating:** 3
**Confidence:** 3

**Summary:**

This paper proposes ZeroSumEval, an LLM dynamic benchmark with a diverse suite of games to measure the abilities of LLM models. The target abilities claimed for measurement include reasoning, planning, knowledge application, and creativity. The provided and designed games are the board game, chess, the question-answer game, MathQuiz, and the cybersecurity game, PyJail. The authors report the evaluation ratings (based on the Bradley-Terry model) of four renowned LLM chatbots, including GPT-4o, Claude 3.5 Sonnet, Mistral-Large, and LLaMA 3.1 70B, and present their results. The key idea of this paper is a workflow with automatic prompting to generate the evaluation challenges rather than using static question pools and also includes a verification mechanism to alleviate answer memorization, ensuring that challenges already memorized by LLM models are excluded from the evaluation.

**Strengths:**

* The survey and narrative are well-executed, and the authors effectively highlight many key problems in current LLM evaluation.
* The workflow appears reasonable for conducting this type of dynamic evaluation. I believe dynamic evaluation can mitigate some of the current weaknesses of static evaluation, and prebuilt games with clear rules can serve as an alternative solution to costly and potentially biased human evaluations.
* The attached codes seem to help readers reproduce and use this benchmark.

**Weaknesses:**

* Although the motivation is good and the problem is clear, the proposed method remains a gap to what the authors described as the current evaluation problem. For example, it is not clear how ZeroSumEval can address the data contamination problem, especially as it still uses existing game rules without any rephrase techniques in reasoning ability. Some recent LLM research shows that LLMs may not have genuine reasoning abilities when symbolic elements in prompts are rephrased, and the results may not be as desired. Why are there no mechanisms for replacing pieces like king, queen, or knight with P1, P2, P3, etc., to prevent pattern memorization that masks real reasoning and planning ability? An agent equipped with true reasoning and planning ability should easily adapt to symbolic changes but with identical rules.

* For purely automatic prompt generation, how can the quality of these prompts be guaranteed without human oversight? While the goal includes reducing human bias, the proposed method seems to remain susceptible to some bias. No mechanism is described, such as using an ensemble of different LLMs to achieve better consensus.

* Moreover, the evidence of creativity is weak. The authors do not even define what constitutes "creativity," such as decisions that differ from known results but are still reasonable. I cannot be convinced by merely analyzing word distribution, as it is not a clear and well-defined measure for creativity.

* For an LLM evaluation, rigor is essential since the score will represent the ability of LLM models. I found the methodology and experimental evidence too weak to support this paper, falling short of the standards expected at ICLR.

**Questions:**

* You stated in Section 3.3, "The design also correctly penalizes models that directly generate memorized questions as it is likely to have been memorized by other models, thereby encouraging models to create challenging and novel questions." What is the actual verification mechanism you used to prevent memorization? Can this mechanism truly alleviate question or answer memorization?

* How do you guarantee the quality of automatic prompting in Section 3.4 without human oversight, especially considering bias?

* What is the value of the reported rating from the BT model? Is it simply $\frac{r^1}{r^1+r^2}$, or does it involve an exponential reparameterization or other common transformations to align with Elo rating interpretations?

* Scalar rating systems like the BT model cannot handle cyclic dominance or win-value intransitivity. How do you ensure that your generated challenges reflect balanced and fair scoring? Additionally, what diversity measures do you use in challenge generation to ensure coverage of questions rather than similar series types that are not identical?

---

### Official Review · Reviewer_qno3 · 2024-11-08

**Soundness:** 1
**Presentation:** 2
**Contribution:** 1
**Rating:** 3
**Confidence:** 3

**Summary:**

This paper introduces ZeroSumEval, a framework for evaluating LLMs through zero-sum games: the work uses chess, writing and solving math problems, and CTF tasks. The approach touts to address benchmark saturation, prompt sensitivity, and the high costs of human evaluation.

**Strengths:**

The paper focuses on an important topic: how to make benchmarks for LLMs that do not easily saturate, and isn't sensitive to details such as prompts that may affect models differentially in way which may bias evaluation?

As far as I'm aware, the idea of using zero-sum games is original. That being said, having "win rates" between models as an evaluation scheme is quite common, which seems similar to setting up evaluation in a zero-sum manner. I'm not too familiar with works that do this, but would be useful to understand how this differs from that.

I also found the scheme for "scalable verification" to be clever, even though it seems hard to verify whether it would hit edge cases once models are sufficiently capable.

**Weaknesses:**

Broadly, the paper seems like an incomplete submission in ways that are substantial enough to warrant rejection.

As far as I can tell, the paper:
1) Is missing crucial details that are fundamental for a thorough evaluation of it:
   1) The paper does not provide examples of the environments, or how and when the scalable verification protocol fails or succeeds. It seems basically impossible to evaluate the paper without such examples.
   2) Relatedly, the code is missing for the paper, despite it being primarily a benchmark paper.
   3) The paper is also missing crucial details about exactly how the prompt optimization scheme is applied, how evaluations are run, etc. For example, the authors say: "We create standard datasets manually for each game available to all models for the optimization. The available datasets are described in Table 1.".  Is this just for prompt optimization? What are example prompts that are found? How are those then used to generate scenarios? What are the prompts used at every point of the process?
   4) The authors remain very vague about details the mention: e.g. "50-100 games per experiment". Is this different for different environments? Within the same environment? Why?
   5) What environment(s) is table 3 discussing?
   6) Despite all the missing information above, which is readily apparent on a first read of the paper, the authors do not use all the space at their disposal for the submission.

2) Makes misleading claims about results, and does not present promised results:
    1) Figure 2 seems to show the opposite of what the authors claim (or at the very least have mixed evidence against what you claim)? "For example, in a chess game, models equipped with optimized prompting demonstrated a higher proportion of correct moves compared to their counterparts using default prompts Figure"
    2) "We find ZeroSumEval correlates strongly with expensive human evaluations (Chatbot Arena) and disagrees with benchmarks with known overfitting and saturation issues." -> I don't think you actually show this anywhere? This seems like an important result
    3) "Gandalf" is in Figure 1, but not present in the paper – curious as to why not? It's normal for benchmark environments to be cherry picked to be appropriate, but the reasons for lack of inclusion for another env may still be informative into the promise of the method.
    4) The effect of prompt optimization methods seems to be quite small, CIs have large overlaps (Table 3), and interpretations such as "Prompt optimization can even flip ranking as is the case with MIPROv2" do not seem to have much weight in light of that.
    5) The suggestion that more words would also relate to deeper reasoning and better performance is also not necessarily supported from the data: e.g. llama3.1-70B shows the most word usage (when non prompt optimized), but performs worse to prompt optimized alternatives.

More broadly, I'm also somewhat confused on a high level of why the authors think that the performance on chess would be indicative of general reasoning capabilities. I think as long as they can demonstrate clear mappings between scores on this benchmark and other benchmarks, that would be convincing, but without it, this benchmark seems somewhat contrived.

I'm also not sure if it's possible for LLMs to prove that math problems are valid and solvable with the procedure that the authors describe. This seems especially challenging when we're dealing with superhuman math (which is kind of what this benchmark is).

**Questions:**

The authors say that the benchmark "measures AI models’ abilities to self-improve from limited observations" -> I'm confused about this statement. What are the limited observations?

---

### Note · Authors · 2024-11-19

I have read and agree with the venue's withdrawal policy on behalf of myself and my co-authors.